# Preparation and Enhanced Antimicrobial Activity of Thymol Immobilized on Different Silica Nanoparticles with Application in Apple Juice

Yuhao Liu [1], Xutao Li [1], Jie Sheng [1], Yuyang Lu [1], Huimin Sun [1], Qixiang Xu [1], Yongheng Zhu [1,2,*] and Yishan Song [1,3,*]

1   College of Food Science and Technology, Shanghai Ocean University, Shanghai 201306, China; lyhns95@163.com (Y.L.); 2021230@st.shou.edu.cn (X.L.); jsheng@shou.edu.cn (J.S.); ankanghou@126.com (Y.L.); tj215@126.com (H.S.); shoufile@126.com (Q.X.)
2   Laboratory of Quality & Safety Risk Assessment for Aquatic Products on Storage and Preservation, College of Food Science and Technology, Ministry of Agriculture and Shanghai Engineering Research Center of Aquatic-Product Processing & Preservation, Shanghai Ocean University, Shanghai 200120, China
3   Department of Chemistry, Shanghai Ocean University, Shanghai 201306, China
*   Correspondence: yh-zhu@shou.edu.cn (Y.Z.); yssong@shou.edu.cn (Y.S.)

**Abstract:** In order to diminish the application limitations of essential oils (EOs) as natural antimicrobial components in the food industry, novel antimicrobial materials were designed and prepared by immobilization of thymol derivatives on silica particles with different morphologies (hollow mesoporous silica nanoparticles, MCM-41, amorphous silica). The structural characteristics of antimicrobial materials were estimated by FESEM, FT-IR, TGA, N2 adsorption-desorption, and small-angle XRD, and the results revealed that both mesoporous silica nanoparticles maintained the orderly structures and had good immobilization yield. Furthermore, the antibacterial performance tests showed that mesoporous silica nanoparticles greatly enhanced the antimicrobial activity of thymol against two representative foodborne bacteria (*Escherichia coli* and *Staphylococcus aureus*), and the application of the antimicrobial support was tested in apple juices inoculated with *E. coli*. The MBC of functionalized mesoporous silica supports was established to be below 0.1 mg/mL against *E. coli* and *S. aureus*, which is much lower than that of free thymol (0.3 mg/mL and 0.5 mg/mL against *E. coli* and *S. aureus*, respectively). In addition, at a range from 0.05 mg/mL to 0.2 mg/mL, immobilized hollow mesoporous silica nanoparticles (HMSNs) can inhibit the growth of *E. coli* in apple juice and maintain good sensory properties during 7 days of storage.

**Keywords:** silica nanoparticles; mesoporous structure; antimicrobial activity; immobilization; thymol

## 1. Introduction

Food-borne diseases caused by contaminated food are a serious threat to public health and a huge impediment to socio-economic development worldwide [1]. Typically, apple juice is generally considered a low-risk food due to its low pH, reducing the possibility of bacterial survival. However, it has been found that apple juice was related to multiple foodborne illness outbreaks caused by *E. coli* [2]. Since many foodborne diseases are caused by ingestion of food contaminated by microbial pathogens, many studies are focused on counteracting the action of such food-related pathogens [3]. Moreover, the emergence of microbial drug resistance due to the abuse of antibiotics and consumers' preference for eco-friendly food preservatives has promoted the rapid and innovative development of food product technology [4,5]. Some new tendencies in this field include the use of naturally-occurring antimicrobial compounds, such as essential oils (EOs) [6]. EOs have been proved to be good sources of bioactive compounds with antimicrobial properties [7]. Several EOs have been registered by the European Commission for use as flavorings in

foodstuffs and classified as GRAS (generally recognized as safe) by the FDA (the US Food and Drug Administration) due to the lack of health risks to the consumer [8,9]. Furthermore, some studies have confirmed that one of the most effective antibacterial components in EOs is coming from the thymus (thymol) [10]. Thymol (2-isopropyl-5-methylphenol) is the main monoterpene phenol occurring in EOs and has been used in the food industry since ancient times as flavoring and preservative agents, thanks to its antimicrobial properties [8]. Given an excellent antimicrobial function, thymol undoubtedly has high potential in the food industry as natural additives [11]. However, its hydrophobicity makes it insoluble in the food matrix and easier to combine with lipids to reduce antimicrobial activity, which will limit its application in food. Moreover, it has strong sensory properties, high volatility, and degradability, which will adversely affect the sensory properties of foods upon incorporation in the free form or at high concentrations [10,12–14]. Nevertheless, the recent advancement in nanotechnology offers new strategies for the EOs to reduce the adverse effects on sensory properties with improving antimicrobial potency in the food matrix, such as encapsulation or immobilization [9,15,16].

Recently, serials studies have been devoted to improving EOs' antibacterial activity by drug delivery systems using different organic nanomaterials based on solid-lipids, nanoemulsions, and liposomes [17–19]. However, these organic delivery systems have some limitations such as physical and chemical stability, capsule compatibility, high cost of production, and the lack of food-grade coating materials [9].

Besides traditional organic matrices, inorganic formulations are now considered the preferential choice. For example, clay hybrids allow high stability and controlled release of the essential oil compounds [20]. Especially, mesoporous silica particles have been widely proposed as delivery systems in medicine and food technology in recent years thanks to their unique features, such as stability, biocompatibility, and large load capacity [21–23]. Among them, hollow mesoporous silica particles (HMSNs) can efficiently encapsulate drugs within their hollow cavity, making them one of the most promising nanocarriers for hydrophobic compounds [24,25]. At the same time, MCM-41 is also the most widely used porous silica in applications in the food sector [26]. Besides mesoporous silica particles, other silica particles are gradually used in the food industry. For example, synthetic amorphous silica (AS) has been used as a direct food additive for decades and has been proved to have no risk to human health [27,28]. Apart from entrapping drugs, siliceous materials present a large surface that can be easily modified with various functional molecules [23]. Based on this approach, Ruiz-Rico et al. [6] reported the antimicrobial activity of several EOs grafted onto the surface of three silica supports with different morphologies. Sokolik et al. [29] evaluated the antibacterial activity of the carvacrol-containing hybrid silica against *E. coli*. To the best of our knowledge, there have been several studies which have researched improving the antimicrobial activity of EOs by anchoring it onto the surface of silica supports, however, the antimicrobial activity of EOs immobilized on HMSNs has not been studied and reported yet.

In this scenario, we previously reported the synthesis and antibacterial properties of thymol-functionalized HMSNs supports [30]. As a step forward, thymol was grafted on the surface of three types of silica particles (HMSNs, MCM-41, AS). The synthesis was conducted in a two-step process, in which antibacterial supports were synthesized after the synthesis of thymol derivatives. Then, the characterizations of bare and functionalized silica supports were conducted by FESEM, FT-IR, TGA, and small-angle XRD. Additionally, contrast to free thymol, the antimicrobial activity of functionalized silica particles against *E. coli* and *S. aureus* was evaluated, and the inhibition effect of HMSNs on the growth of *E. coli* in a real food system was also studied.

## 2. Materials and Methods

### 2.1. Reagents

2-isopropyl-5-methylphenol (Thymol, 98%), tetraethyl orthosilicate (TEOS, 98%), aqueous ammonia (28%), cetyltrimethylammonium bromide (CTAB, 99%), absolute ethanol

(99.8%), dimethyl sulfoxide (DMSO, 99%), sodium and chloride (NaCl, 99%), and micro-biological media were provided by Sangon Biotech (Shanghai, China). 3-(triethoxysilyl)-propyl-isocyanate (TEPIC, 95%) and tetrahydrofuran (THF, 99%) were obtained from Al-addin Biochemical Technology Co., Ltd. (Shanghai, China). The apple juice was purchased from the local supermarket.

### 2.2. Silica Nanoparticles and Thymol Derivatives Preparation

A one-step method was used to synthesize the antimicrobial supports. Firstly, the thymol was reacted with 3-(triethoxysilyl)-propyl-isocyanate (TEPIC) to yield the corresponding alkoxysilane derivatives, thymol-3-(triethoxysilyl)-propyl-isocyanate (TTEPIC). Then, TTEPIC was directly added in different reaction systems to react with TEOS and different functionalized supports were obtained.

Thymol was first dissolved in a small quantity of tetrahydrofuran solvent (THF). After complete dissolution, TEPIC was dropwise added into the solution with a molar ratio fixed at 1 thymol: 1 TEPIC. The mixture was stirred at 65 °C in a covered flask for approximately 18 h in the nitrogen atmosphere. Finally, the mixture was concentrated at room temperature to remove the solvent THF using a rotary vacuum evaporator and a clear oil was obtained.

The functionalized HMSNs were synthesized according to a modified literature protocol [25]. Briefly, 0.9 g of CTAB was dissolved in 300 g of distilled water and 142 g ethanol. Then, 7.3 g of ammonia was added to the mixture under constant stirring for an hour. Next, 5.6 g of TEOS and 2.2 g of TTEPIC were added dropwise to the solution and stirred at 35 °C for 6 h. Following synthesis, the solid was recovered and washed with deionized water. After being dried at 60 °C overnight, 0.5 g of the solid was added in 250 g of deionized water and heated at 90 °C for 24 h in the water bath to synthesize the hollow structure. Subsequently, the solid was collected with a filter, washed with deionized water and dried under a high vacuum overnight. Finally, the sample was refluxed with ethanol in Soxhlet extraction for 2 days to remove the template, giving the functionalized HMSNs.

Based on the literature [26], functionalized MCM-41 was synthesized as follows: 127 g of ammonia and 2 g of CTAB were added to a round-bottomed flask to be dissolved completely by 200 g of deionized water at 60 °C stirring for 1 h at 250 rpm. Then, 7.5 g of TEOS and 2.9 g of TTEPIC were added dropwise to the solution and the reaction was stirred for 6 h. Following, crystallization was carried out at 33 °C for 24 h and the mixture was filtered, washed, and dried at 50 °C. Finally, the template was removed by refluxing with ethanol in Soxhlet extraction for 2 days, obtaining the functionalized MCM-41.

The functionalized AS was obtained according to the following procedure [28]: 8 g of TEPIC was mixed with 78.9 g of ethanol then stirred for a few minutes. Subsequently, 1.87 g of TEOS and 7.28 g of ammonia were added to the solution and the reaction was stirred for 24 h. Following synthesis, the sample was separated by centrifugation at 12,000 rpm for 20 min at 4 °C, washed with deionized water and ethanol, and dried overnight.

As references for characterization, different types of bare supports were synthesized without adding TTEPIC following the method described above.

### 2.3. Characterization

The characterizations of bare and functionalized silica supports were conducted by standard techniques including field emission scanning electron microscopy (FESEM), Fourier transform infrared spectroscopy (FT-IR), thermogravimetric analysis (TGA), and small-angle X-ray diffraction (XRD).

The morphological structures of silica particles were observed by FE-SEM (Hitachi SU5000, Tokyo, Japan). Powder of dried samples was distributed on double-sided copper conductive tape and gold particles were sprayed onto the sample surface. Finally, FESEM images were obtained at an accelerating voltage of 6 kV.

The chemical composition analysis was performed by FT-IR (Nicolet Instrument, Thermo Company, Waltham, MA, USA) with a spectrometer equipped with a DTGS

detector and a Golden Gate diamond ATR accessory. Spectra were recorded by averaging 64 scans at a resolution of 4 cm$^{-1}$ in the 4000–400 cm$^{-1}$ range.

The functionalized degree of silica particles was determined by TGA (Netzsch STA 409, Germany) using a heating program that consisted of a heating ramp of 10 °C/min from 30 to 800 °C in an oxidant atmosphere (air, 80 mL/min).

The synthesis process and the structure of the mesoporous silica particles were evaluated by small-angle XRD using a Bruker AXS D8 X-ray diffractometer (Karlsruhe, Germany) with an operation voltage of 40 kV and current of 40 mA.

### 2.4. Antibacterial Activity Assays

#### 2.4.1. Culture Conditions and Bacterial Strain

Two strains of *E. coli* (K12D31, Gram-negative) and *S. aureus* (ATCC 27661, Gram-positive) were obtained from Shanghai Ocean University, China. All strains were stored at 4 °C in tryptic soy agar (TSA) before use. The cells from a colony grown on TSA were transferred to 10 mL of tryptic soy broth (TSB) and incubated at 37 °C for 24 h to obtain an inoculum density of approximately 10$^8$ CFU mL$^{-1}$ for testing.

#### 2.4.2. Antibacterial Activity Assays In Vitro

The antimicrobial activity of thymol against E. coli and S. aureus was determined by minimum inhibition concentration (MIC) and minimum bactericidal concentration (MBC) based on a micro-well dilution method [31]. Different amounts of thymol were individually dissolved in 5 % DMSO. TSB was added in sterile 96-well plates and binary dilutions were performed to obtain final concentrations ranging from 0.002 to 4 mg/mL. After performing the binary dilutions, 10 μL of tenfold diluted microbial suspension were added to each well, to provide an initial cell density of approximately 10$^6$ CFU/mL. All microtiter plates were incubated at 37 °C for 24 h. Positive and negative controls were included in all assays. MIC values were defined as the minimal concentration of antimicrobial compound that inhibits visible growth of the strains. MBC was estimated from the same microplates used to determine the MIC. To do this, we spread aliquots from the wells used to estimate the MICs (no evidence of growth) and from wells in the previous rows on plates. MBC was the lowest concentration with no apparent microbial growth on agar after incubation at 37 °C for 24 h. All the treatments were performed in triplicate.

The antimicrobial activity of the functionalized supports was tested using the thymol concentrations. Equivalent amounts of the immobilized thymol were calculated according to the functionalized degree of antimicrobial supports. The different amounts of antimicrobial supports were suspended in 10 mL of TSB in test tubes and inoculated with 100 μL of microbial suspension to provide an initial cell density of approximately 10$^6$ CFU/mL. Then, the test tubes were incubated at 37 °C for 24 h with orbital stirring (150 rpm). Finally, cultivable cell numbers were determined by a serial dilution counting method and incubating at 37 °C for 24 h. The cultivable viable cell numbers were logarithmically converted to log10 CFU/mL. All the treatments were performed in triplicate, including positive control and negative control.

The percentage of cell growth reduction (R, %) was calculated using the following equation:

$$R = (C_0 - C)/C_0 \times 100\% \tag{1}$$

where $C_0$ is the number of CFU from the control sample and C is the number of CFU from treated samples.

#### 2.4.3. Antibacterial Activity Assays in a Real Food System

The antimicrobial activity of the functionalized supports was determined in a real food system by using the apple juice purchased from the local market. According to the results of the antibacterial assay in vitro, functionalized HMSNs and thymol were used to inhibit the growth of *E. coli*, a common pathogen in apple juice. The concentration of thymol was selected based on the MBC results of HMSNs. EquORYvalent concentrations

of the free and immobilized thymol (0.05, 0.1, and 0.2 mg/mL) were added to 10 mL of apple juice and inoculated with $10^3$ CFU/mL of the microorganism. The samples were stored at 4 °C for 7 days to determine the shelf life of refrigerated juice. The total number of bacteria in apple juice was determined on 1, 3, 5, and 7 days. The total number of bacteria was determined by inoculating on TSA with different tenfold serial dilutions of culture medium. After incubation at 37 °C for 24 h, the colony-forming units (CFU) were counted and expressed as log10 CFU/mL. All the treatments were performed in triplicate.

### 2.4.4. Determination of the Physicochemical Properties of the Apple Juice

The physicochemical properties of treated juice during refrigerated storage were determined including color, pH, and soluble solids. The refrigerated juice mixed with HMSNs was filtered to remove the silica particles. The color of the juice was determined by an automatic colorimeter (3nh-ys3060, Shenzhen, China). The pH of the juice was measured by a digital pH meter colorimeter (le438, Mettler, Switzerland). Total soluble solids were determined by a handheld refractometer (Pocket PAL-1) and the results expressed as Brix. The uninoculated juice without being processed was tested as a control.

### 2.5. Statistical Analysis

Data of different groups were collected and analyzed using one-way ANOVA by the SPSS software. The LSD (least significant difference) procedure was utilized to test for differences between averages at the 5% significance level.

## 3. Results and Discussion

### 3.1. Design and Synthesis of the Antimicrobial Supports

The preparation of antimicrobial supports and experimental procedures are shown in Figure 1. Thymol was anchored on the surface of silica particles to prepare the novel antimicrobial supports which can maintain more stable and long-term antimicrobial effects due to the stability of covalent immobilization compared to encapsulation. Three kinds of silica particles were selected as the supports: HMSNs, MCM-41, and AS particles. HMSNs have attracted great interest recently due to their hollow and mesoporous structure which can hold a higher amount of drug compared to traditional mesoporous silica [32]. Besides, MCM-41 is another mesoporous silica with hexagonal arrays of cylindrical mesopores and has been extensively studied in past years [33]. Finally, AS particles are non-crystalline structures of silicon dioxide widely used in the medical field due to their high biocompatibility [27].

Post-grafting is considered a common method used to immobilize EOs on the surface of silica particles, which means that the supports need to be prepared before modification [6]. In contrast, a simpler and easier method designated as one-step synthesis was adopted to synthesize different antimicrobial supports in this study. In this method, organic molecules were covalently anchored on the surface of supports through the co-condensation of TTEPIC and TEOS in the presence of the structure directing agent (CTAB). Compared with the post-grafting method, this method can not only simplify the synthesis procedure to reduce the generation of intermediate products and improve the purity of the product, but also make the organic molecules more homogeneously distributed due to the co-condensation process [34].

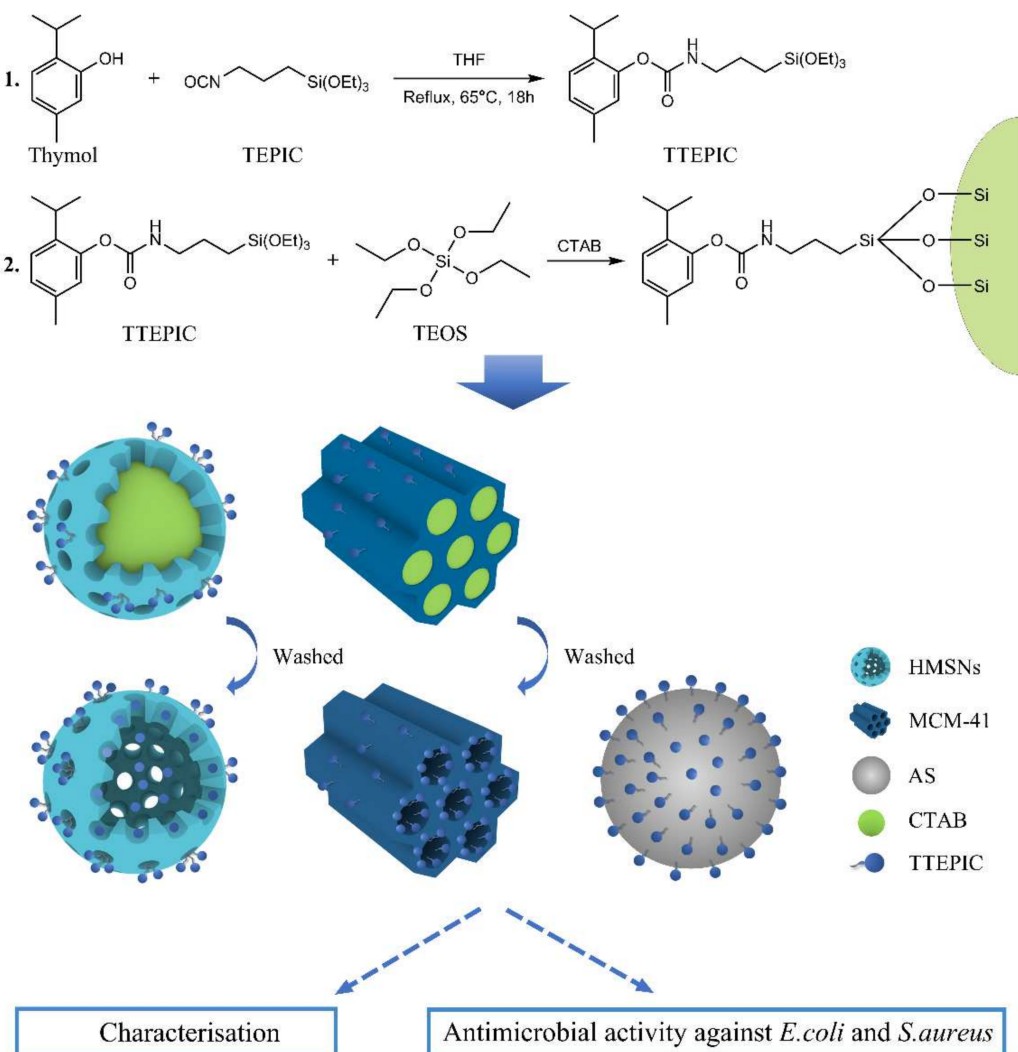

**Figure 1.** Schematic representation of the experimental procedure.

### 3.2. Characterization of Antimicrobial Supports

The particle size and morphology of the bare and functionalized supports were observed by FESEM in Figure 2; while bare and functionalized HMSNs particles (Figure 2A,B) showed the unique and uniform structure of hollow sphere with diameters ca. 300 nm, which was similar to the previously reported [25]. As shown in Figure 2C,D, the MCM-41 particles presented hexagonal morphology with particle size in the microscale range (reported particle size of ca. 4 μm) [6]. It is also worthwhile noting that no changes in the structure of the different mesoporous supports were detected when comparing the bare and functionalized samples, which confirms the immobilization procedure does not alter the integrity of the mesoporous silica particles. However, the size of AS particles has been changed after functionalization shown in Figure 2E,F. The diameter of the functionalized particles was much smaller than that of the bare particles, although both of them presented a roughly spherical structure. This may be due to TTEPIC, especially the isopropyl group, causing steric hindrance of the reactive sites for condensation that prevents further particle growth [29].

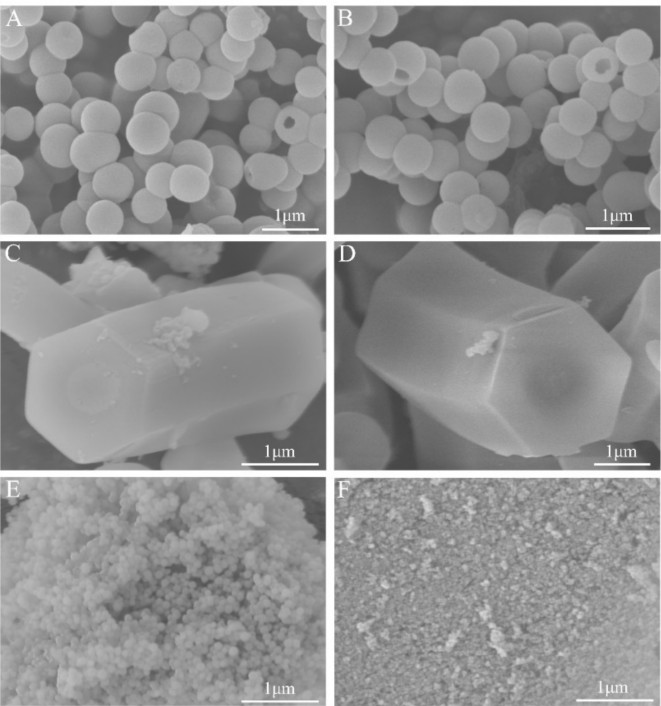

**Figure 2.** FESEM image of the bare and functionalized HMSNs (**A**,**B**), MCM-41 (**C**,**D**), and AS (**E**,**F**).

Figure 3 depicts the FT-IR spectra of the different bare, functionalized supports, and TTEPIC. In general, the different bare and functionalized supports exhibited a large and unresolved envelope located in the region 1300–1000 cm$^{-1}$, and medium bands at 791 cm$^{-1}$ and 460 cm$^{-1}$, which are characteristic absorption bands of silica respectively assigned to the asymmetric stretching vibrations of Si-O-Si, symmetric stretching vibration of Si-OH, and symmetric stretching vibrations of Si-O [35–37]. The weak peaks at 2927 cm$^{-1}$ and 2856 cm$^{-1}$ appeared in mesoporous silica particles (Figure 3(Aa–Ad)) and can be attributed to the antisymmetric and symmetric stretching modes of CH$_2$ in CTAB, respectively [24,35]. In addition, the presence of a broad band from 3600 to 3000 cm−1 in the functionalized particles and TTEPIC (Figure 3(Ab,Ad,Af),B) corresponded to the O-H bonding vibration of adsorbed water and N-H bonds of TTEPIC, and peaks' range at 1600–1420 cm−1 was as-signed to the C=C stretching of the aromatic ring [36,38,39]. Unfortunately, the C-O-C vibration peak of TTEPIC was covered by the vibration of Si-O-Si, although the mentioned evidence enough confirmed the presence of TTEPIC on the surface of functionalized silica particles.

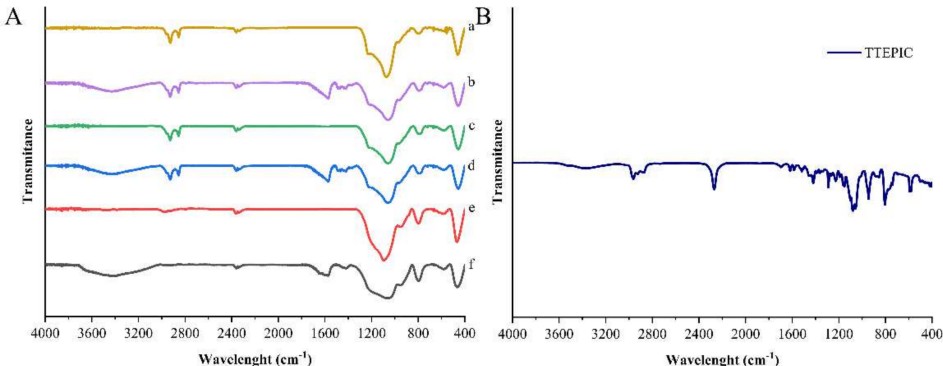

**Figure 3.** FT-IR spectra of the bare and functionalized HMSNs (**Aa**,**Ab**), MCM-41 (**Ac**,**Ad**), AS ((**Ae**,**Af**), Karlsruhef) and TTEPIC (**B**).

Figure 4A shows the TGA curves of bare and functionalized supports. The different bare supports showed a one-step of weight loss in the range of 30–180 °C, which can be attributed to the loss of absorbed water molecules and used solvent in the synthetic procedure. The TGA curve of functionalized particles exhibited a two-step weight loss. Compared with bare particles, the preliminary stage before 180 °C showed less weight loss. During the next stage (200–800 °C), the weight loss was attributed to the decomposition of the organic moiety on the surface of the supports, which proved thymol was successfully grafted on the surface of silica particles. Meanwhile, these results showed that the amount of thymol grafted on the surface of different supports was about 7.11% (HMSNs), 7.36% (MCM-41), and 10.39% (AS), respectively. In addition, thymol was completely degraded before 150 °C shown in Figure 4B, which indicated that functionalization effectively improved the stability of thymol. Similarly, the organic moiety of TTEPIC is also rapidly degraded during 30–150 °C.

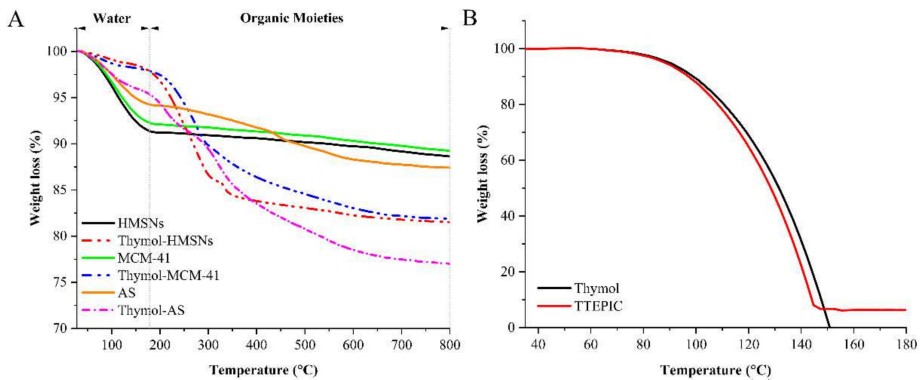

**Figure 4.** TGA curves of the bare and functionalized supports (**A**), thymol and TTEPIC (**B**).

In Figure 5, small-angle X-ray diffraction (XRD) patterns of the host matrix MCM-41 and the functional antibacterial support are shown. It can be seen that MCM-41 exhibited characteristic diffraction peaks at 2θ = 2.09°, 3.64°, and 4.18°, ascribable to the (100), (110), and (200) atomic planes associated with two-dimensional cylindrical pores arranged in p6 mm hexagonal symmetry [40,41], representing a typical MCM-41 XRD pattern. Compared to MCM-41, the diffraction intensities of Thymol-MCM-41 is 2.09°, implying that the main structure of mesoporous silica was not altered. However, Thymol-MCM-41 exhibited no obvious characteristic peak at 3.64° and 4.18°, suggesting that thymol was well introduced on the surface of mesoporous silica, and the characteristic peak cannot be detected by X-ray because the electrostatic coating of thymol slightly reduced the order of the mesoporous silica. The results are consistent with the results of FESEM.

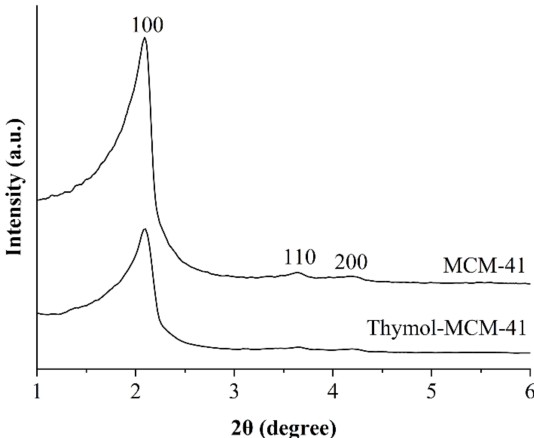

**Figure 5.** Small-angle XRD patterns of the bare and functionalized MCM-41.

### 3.3. Antimicrobial Activity of Antimicrobial Supports In Vitro

The antimicrobial activity of the free thymol and antimicrobial supports was established by determining the MIC and MBC of two representative food-borne microorganisms: *E. coli* and *S. aureus* [42]. Figure 6 presents the growth reduction of *E. coli* and *S. aureus* treated with free thymol and different antimicrobial supports for 24 h.

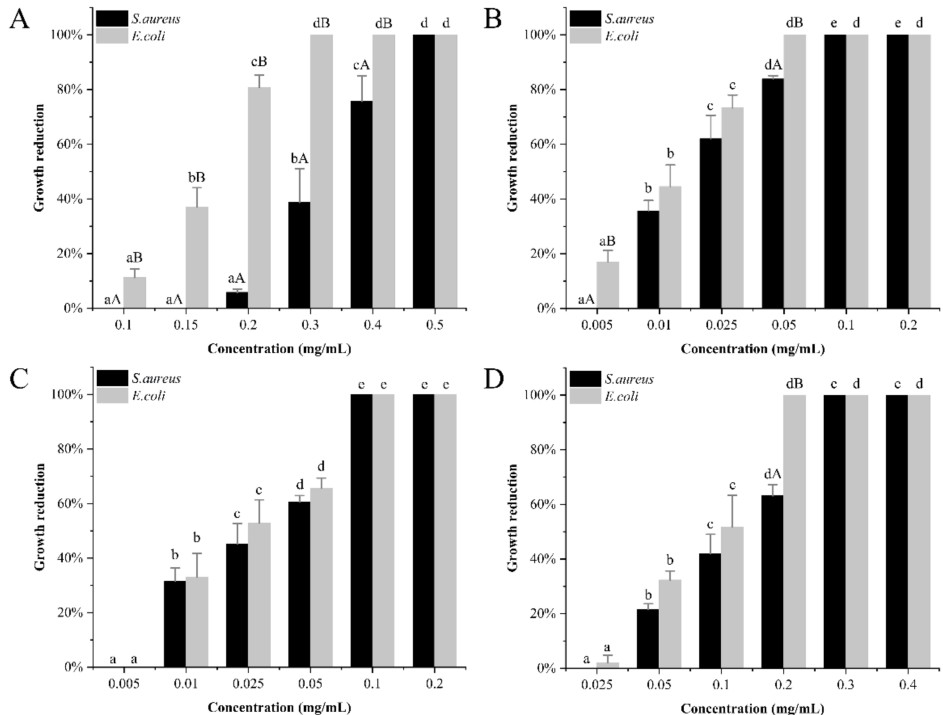

**Figure 6.** The growth reduction of *E. coli* and *S. aureus* treated with free thymol (**A**), functionalized HMSNs (**B**), MCM-41 (**C**), and AS (**D**). Different letters in the bars indicate statistically significant differences ($p < 0.05$) from concentrations (small letters) and microorganisms (capital letters) (n = 3).

The effect of free thymol on the growth of *E. coli* and *S. aureus* is shown in Figure 6A. The growth of *E. coli* was completely inhibited at a range from 0.3 mg/mL to 0.5 mg/mL and partial inhibition of microorganisms was observed at concentrations between 0.1 mg/mL and 0.2 mg/mL. The MIC and MBC of free thymol against *E. coli* was at 0.3 mg/mL, which is similar to the previous studies [43,44]. According to the different degrees of effect on the growth of *S. aureus*, different concentrations can be divided into three groups: complete inhibition (0.5 mg/mL), partial inhibition (0.2–0.4 mg/mL), and no reduction (0.1–0.15 mg/mL). The value of MIC and MBC for *S. aureus* at 0.5 mg/mL is consistent with the study of Rua, Fernandez-Alvarez [45]. Compared to *S. aureus*, free thymol exhibited a more effective inhibition effect on *E. coli*. In particular, the antibacterial activity of thymol against the two bacteria showed significant differences at a concentration range of 0.1–0.4 mg/mL, and the difference was gradually reduced with the increased concentration. These results suggested the greater antimicrobial activity of thymol against *E. coli*, which are in accordance with the study of Gutiérrez-Larraínzar, Rúa [44].

As shown in Figure 6B, the antimicrobial activity of thymol was remarkably enhanced by functionalized HMSNs. The growth of *E. coli* was partially inhibited by 0.005–0.025 mg/mL concentration. The value of MIC and MBC was reduced to 0.05 mg/mL, which is equivalent to one-fiftieth of free thymol. Similarly, the value of MIC and MBC for functionalized HMSNs against *S. aureus* (0.1 mg/mL) was also much lower than that of free thymol. Besides, compared to the free component, the growth reduction of *S. aureus* treated with functionalized HMSNs was closer to the *E. coli* at partial inhibition concentration, which indicates that *S. aureus* is more susceptible to the thymol grafted on the silica supports.

In fact, the same trend can also be observed in the functionalized MCM-41 shown in Figure 6C. The MIC and MBC of functionalized MCM-41 for *E. coli* and *S. aureus* were both at 0.1 mg/mL and the antibacterial activity of functionalized MCM-41 against the two bacteria showed no significant difference ranging from 0.05 mg/mL to 0.2 mg/mL. Nevertheless, at the same concentration of thymol, the bactericidal efficiency of functionalized MCM-41 against the two bacteria (especially against *E. coli*) was slightly lower than that of functionalized HMSNs.

The bactericidal effect of functionalized AS is shown in Figure 6D. The values of MIC and MBC against *E. coli* and *S. aureus* are 0.2 mg/mL and 0.3 mg/mL, respectively. In reference to these values, the enhancement of the antibacterial activity of thymol immobilized on AS is not obvious compared to functionalized mesoporous particles. However, it is worth noting that low concentrations (0.025–0.2 mg/mL) of immobilized thymol on the AS supports displayed improved antimicrobial activity compared to the free compound.

These results indicated that functionalized silica supports not only enhanced the antimicrobial activity of thymol but also improved susceptibility of Gram-positive bacteria to thymol, especially in functionalized mesoporous silica particles. The immobilization of EOs on the surface of silica particles to improve their antimicrobial activity has been reported in few articles previously. Sokolik and Lellouche [29] used a one-step synthesis method to prepare carvacrol hybrid amorphous silica and evaluated its antimicrobial activity against *E. coli*. However, the results showed that the MBC of carvacrol contained in silica particles was 1.4 mg/mL, which is higher than the MBC of free carvacrol (0.35 mg/mL). Conversely, Ruiz-Rico, Perez-Esteve [6] prepared different silica particles (MCM-41, fumed silica, and amorphous silica) functionalized with several EOs by post-grafting method and confirmed their enhancement for the antimicrobial activity of EOs against *E. coli* and *L. innocua*. Among them, thymol-functionalized MCM-41 and amorphous silica exhibited great improvement for the antimicrobial activity of thymol against *E. coli*, which was also confirmed in this study. However, in the current study, the thymol- functionalized MCM-41 showed much higher antimicrobial activity compared with the reported research, which can be due to the advantage of the one-step synthesis method leading to organic molecules more homogeneously distributed than post-grafting.

The effective antimicrobial activity of functionalized supports is based on the activity of thymol derivative. The thymol was reacted with the coupling agent to generate the corresponding derivative, and the Si-OCH$_2$CH$_3$ bond of the derivative was easily hydrolyzed and reacted with TEOS to generate Si-O-Si, and finally, the corresponding functionalized mesoporous silica particles were formed. Bacterial death can be related to exposure to thymol derivatives immobilized on the supports. In addition, different morphology, particle size, and degree of functionalization will affect the antibacterial activity of antibacterial supports [6]. The most effective bactericidal activity of functionalized HMSNs can be attributed to the small particle size and hollow mesoporous spherical structure, which increased the probability of a particle coming into contact with bacterial cells [6].

### 3.4. Antimicrobial Activity of Antimicrobial Supports in Apple Juice

Apple juice has been always considered a low-risk food because of the low pH. However, the acid resistance of *E. coli* gives it the ability to survive in low temperatures and pH conditions, resulting in the outbreak of related foodborne diseases [2,46]. Therefore, the antimicrobial activity of different concentrations of functionalized HMSNs and free thymol against *E. coli* in the apple juice at 4 °C was evaluated and shown in Figure 7. The concentrations of HMSNs were based on the lowest three concentrations that could completely kill bacteria at 0.05 mg/mL, 0.1 mg/mL, and 0.2 mg/mL according to the previous results. The concentration of thymol was set to the same concentration as HMSNs and was used as a control. Compared with the control group, 0.05 mg/mL concentration of free thymol in the apple juice exhibited no inhibition effect. As the concentration increased, 0.1 mg/mL concentration of thymol showed a slight effect on *E. coli* in the first two days. At the concentration of 0.2 mg/mL, the growth of *E. coli* was partially inhibited on the

1 days and completely inhibited on the 3, 5, and 7 days. However, the functionalized HMSNs with equivalent concentrations of thymol completely inhibited the growth of *E. coli* at all concentrations for 7 days, which indicates that the excellent antibacterial ability of functionalized HMSNs was still maintained in a food system.

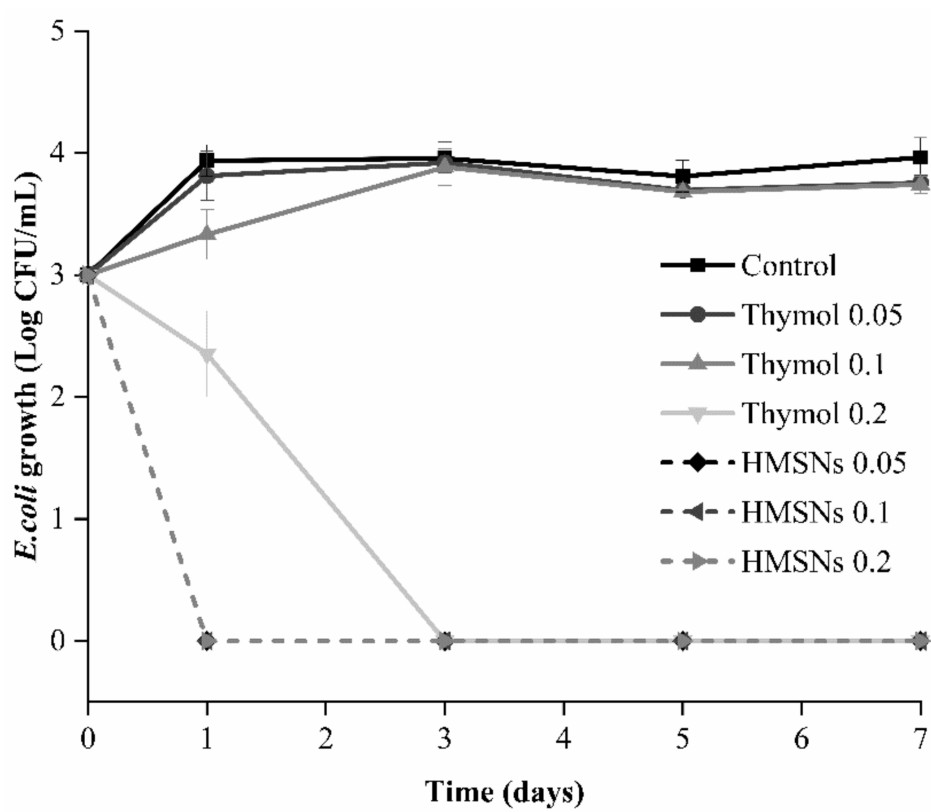

**Figure 7.** Growth of *E. coli* in apple juice treated with different concentrations of free thymol (0.05, 0.1, 0.2) and thymol immobilized on HMSNs (0.05, 0.1, 0.2) during 7 storage days at 4 °C.

### 3.5. Physicochemical Properties of Apple Juice during Storage

The color of apple juice treated with different concentrations of functionalized HMSNs was expressed by parameters (L, a, b) shown in Figure 8A–C. During 7 days of storage at 4 °C, the color of apple juice had little difference between treated and untreated juice. Similar results were obtained in pH and soluble solids of juice shown in Figure 8D,E, which indicated that physicochemical properties of apple juice were hardly influenced by functionalized silica particles. Peña-Gómez, Ruiz-Rico [46] used EOs- (eugenol and vanillin) functionalized silica particle as filtering aids to sterilize apple juice and evaluated the influence of the filtration process. Results showed that the treatment resulted in a slight change in the color and pH of the juice. These results are different from the current study, which can be attributed to the difference in processing methods and functional methods of the silica particle. Conversely, traditional thermal sterilization has been reported to significantly change the color and pH of juice during the thermal process, which will reduce consumer acceptance of fruit juices [47,48]. Therefore, EOs-functionalized silica particles as an alternative to thermal sterilization can effectively reduce the impact on the physicochemical properties of apple juice.

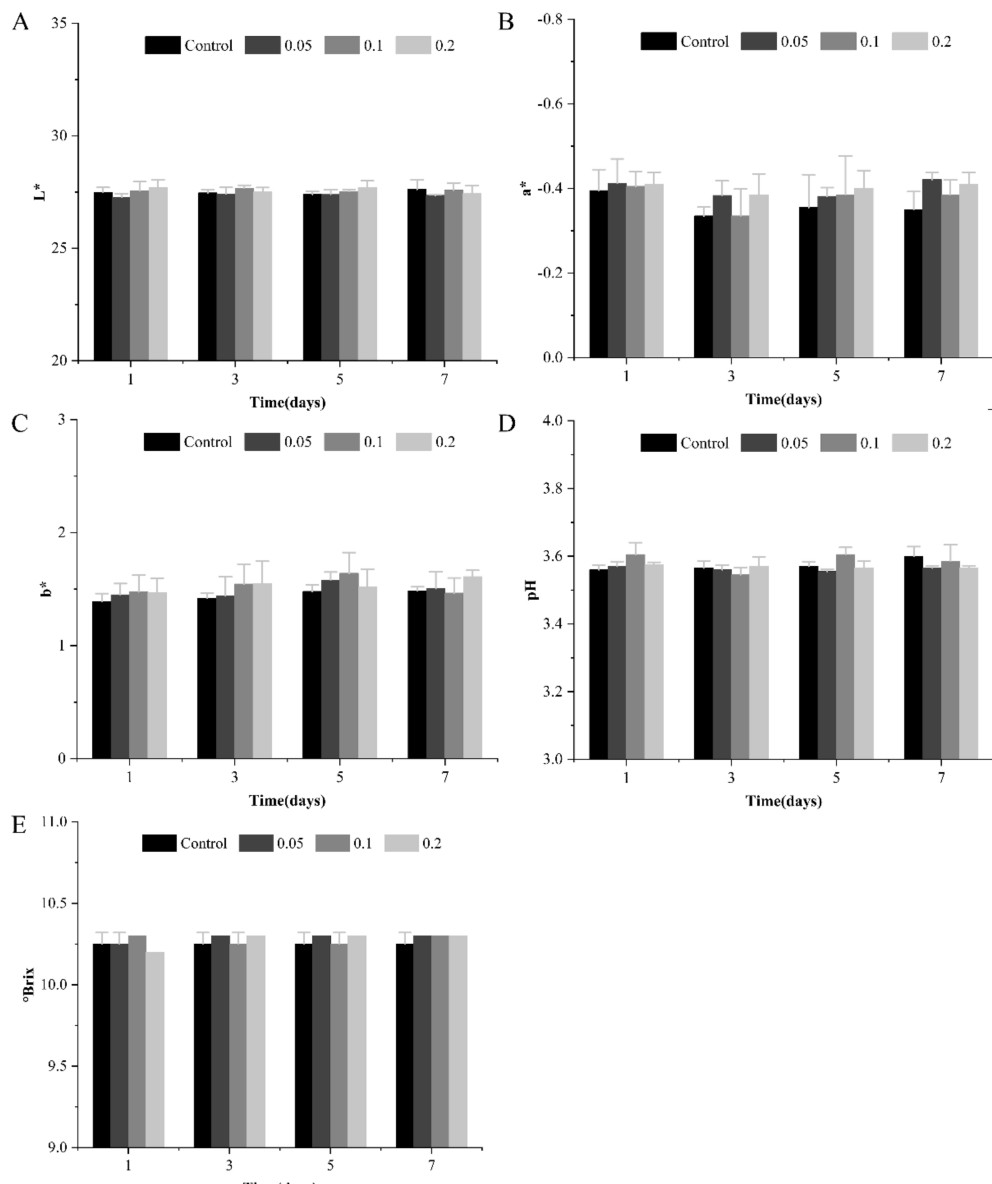

**Figure 8.** The effect of different concentrations of functionalized HMSNs (0.05, 0.1, 0.2) on the color (**A**–**C**), pH (**D**), and soluble solids (**E**) of apple juice inoculated with *E. coli* during 7 storage days at 4 °C.

## 4. Conclusions

In this study, different antimicrobial supports were synthesized based on grafting thymol onto the surface of silica particles with different sizes, morphology, and structure. Three different antimicrobial supports were characterized and proved that the stable structure and regular morphology of the supports have not been altered by the functionalization. The functionalized supports greatly enhanced the antimicrobial activity of thymol against two representative foodborne bacteria (*E. coli* and *S. aureus*). In particular, functionalized HMSNs not only exhibited the highest antibacterial activity but also maintained the effective inhibition effect on the growth of *E. coli* in apple juice due to their unique hollow porous structure. Moreover, the evaluation of the physical and chemical properties of the treated juice proved that functionalized HMSNs hardly affect the quality of the juice. Therefore, EOs immobilized on silica particles have a great potential to improve the antimicrobial activity of EOs and diminish their current limitations in the food industry.

**Author Contributions:** Conceptualization, Y.S., methodology, Y.S. and J.S.; software, Y.L. (Yuhao Liu), H.S. and Q.X.; validation, Y.Z. and Y.S., formal analysis, Y.L. (Yuhao Liu); investigation, Y.L. (Yuhao Liu), Y.L. (Yuyang Lu), H.S. and Q.X.; resources, Y.Z.; data curation, Y.L. (Yuhao Liu) and X.L.; writing—original draft preparation, Y.L. (Yuhao Liu) and X.L.; writing—review and editing, Y.Z. and Y.S.; visualization, Y.L. (Yuyang Lu) and X.L.; supervision, Y.Z. and Y.S.; project administration, Y.Z. and Y.S.; funding acquisition, Y.Z. All authors have read and agreed to the published version of the manuscript.

**Funding:** This research was funded by the Key Research Projects of Science and Technology for Agriculture of Shanghai (2021-02-08-00-12-F00763) of Shanghai Agriculture and Rural Committee.

**Institutional Review Board Statement:** Not applicable.

**Informed Consent Statement:** Not applicable.

**Data Availability Statement:** The data presented in this study are available on request from the corresponding author.

**Conflicts of Interest:** The authors declare no conflict of interest.

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
