# Peer review of "Preparation and Enhanced Antimicrobial Activity of Thymol Immobilized on Different Silica Nanoparticles with Application in Apple Juice"

_coatings, doi:10.3390/coatings12050671_

Round 1

Reviewer 1 Report

Manuscript 16339022

Journal Coatings

Title Preparation and enhanced antimicrobial activity of thymol immobilized on different silica nanoparticles with application in apple juice

The manuscript entitled “Preparation and enhanced antimicrobial activity of thymol immobilized on different silica nanoparticles with application in apple juice” describes the preparation, characterization and food application of a clay hybrid (silica) loaded with thymol derivatives. The topic is not novel. The paper needs substantial revision since the loading capacity and release kinetic of the essential oil compound is not reported, and a general re-organization is necessary. Moreover, the peer-review is very difficult due to the absence of line numbering. The reviewer prepared a new manuscript with line numbering (the file is available at the end of the peer-review report). Please refer to this to follow the comments. Please pay attention to this aspect for future submissions.

Comments are reported in the file.

Author Response

Thank you for your letter and for the reviewers’ comments concerning our manuscript entitled "Preparation and enhanced antimicrobial activity of thymol immobilized on different silica nanoparticles with application in apple juice". Those comments are all valuable and very helpful for revising and improving our paper, as well as the important guiding significance to our researches. We have studied comments carefully and have made correction which we hope meet with approval. Revised portion are marked in yellow in the paper. The main corrections in the paper and responses to the reviewers’ comments are in the attachment.

Reviewer 2 Report

The microbiological part of the manuscript is well structured, so from this point of view the article can be accepted for publication, after a few minor correction:

  • please check the capital letter for Gram positive/negative in the whole document, and also the italics for the bacterial species
  • material and method section
    • the authors should mention if they used ATCC strains or clinical isolates
    • the authors should mention if they used TSB2X as a culture medium for antibacterial assay; if not, the serial dilutions were performed in TSB?
    • line 169 - "for the" should replace "against"
    • all the experiments were performed in triplicate - please check the whole document
    • line 181 - serial dilution counting method?
  • results
    • line 316 - inhibition instead of inhibitory
  • conclusions
    •  the authors mention the in vivo antibacterial activity, but they described only in vitro experiments

Author Response

Thank you for your letter and for the reviewers’ comments concerning our manuscript entitled "Preparation and enhanced antimicrobial activity of thymol derivatives immobilized on different silica nanoparticles with application in apple juice". Those comments are all valuable and very helpful for revising and improving our paper, as well as the important guiding significance to our researches. We have studied comments carefully and have made correction which we hope meet with approval. Revised portion are marked in yellow in the paper. The main corrections in the paper and responses to the reviewers’ comments are in the attachment.

Point 1: please check the capital letter for Gram positive/negative in the whole document, and also the italics for the bacterial species

 Response 1: We have revised this part according to the Reviewer’s suggestion.

Point 2: the authors should mention if they used ATCC strains or clinical isolates

Response 2: We have revised this part according to the Reviewer’s suggestion.

Point 3: the authors should mention if they used TSB2X as a culture medium for antibacterial assay; if not, the serial dilutions were performed in TSB?

Response 3: The serial dilutions were performed in TSB.

Point 4: line 169 - "for the" should replace "against"

Response 4: We have revised this part according to the Reviewer’s suggestion.

Point 5: all the experiments were performed in triplicate - please check the whole document

Response 5: We have revised this part according to the Reviewer’s suggestion.

Point 6: line 181 - serial dilution counting method?

Response 6: Serial dilution coating method.

Point 7: line 316 - inhibition instead of inhibitory

Response 7: We have revised this part according to the Reviewer’s suggestion.

Point 8: the authors mention the in vivo antibacterial activity, but they described only in vitro experiments

Response 8: We have revised this part according to the Reviewer’s suggestion.

Reviewer 3 Report

Dear Author

The manuscript “Preparation and enhanced antimicrobial activity of thymol immobilized on different silica nanoparticles with application in apple juice” “Coatings-1639022” is required to address the following queries

The manuscript emphasize on the thymol immobilization on different silica support systems to improve the bactericidal activity for food preservative particularly apple juice.

Comment 1

This approach is converting “thymol” a GRAS material as per FDA to non GRAS HMSNs formulation, how author will justify it?

Comment 2

Immobilization approach is utilizing, Silica and CTAB which are considered to be toxic “Silica can cause silicosis, whereas CTAB is well known surfactant that can cause protein precipitation and aggregation hence regular consumptions of these two materials in lesser quantities could be harmful, what is author’s opinion about that?  

Comment 3

In the abstract section

Replace the sentence ....the results revealed (from line 23 to 25) .....but also in food system

With

The results revealed that thymol immobilized hollow mesoporus silica particles showed the...... fold higher bactericidal activity against gram negative E. coli and .....fold higher activity against gram positive S. Aureus bacteria  as compared to free thymol.

Comment 4

Page 2. Line 53, 63 –follow the same pattern for reference insertion [ either in bracket or without bracket] 

Comment 5.

Page 2. Line 74  Write the scientist name with et al instead the 6 reported (Ruiz-Rico et al reported )

Comment 6.

Page 2. Line 74-77, reframe the sentence it’s not meaningful in the current from

Comment 7.

Page 3. Section 2.2.2

Include the units of measurements (wt/wt or wt/v or v/v) in the methodology sections of manuscripts

Comment 8.

Page 5. Figure 1. Reaction scheme 1 and 2 is not clear ..insert a high resolution reaction scheme

Comment 9.

Page 6. Line 249

This may be due to TTESPC, Recheck the TTESPC... is it TTEPIC?..Author is advised to go through the entire manuscript carefully for typological errors

Comment 10.

Page 6. Line 266, 267 -Not agree with authors statement, absence of C-O-C peaks could also be result of absence of reaction of thymol and TEPIC at step 1 so include the FT-IR spectra of TTEPIC in figure 3 which will be helpful for comparative analysis of bare HMNS and immobilized HMNS with TTEPIC.

Comment 11.

Page 7. Figure 4 A, include the TTEPIC TGA spectra for comparative analysis

Comment 12.

Page 10. Figure 7. Abbreviation “I” before concentration i.e. I0.05, I0.1,IO.2 is misleading and concentrations is appears to be ten point zero five and so on ....author is advised to change the abbreviation  in figure and figure legend and elsewhere in the manuscript

Comment 13.

Page 11. Figure 8. Abbreviation “I” before concentration i.e. I0.05, I0.1,IO.2 in figure legends is misleading author s advised to change the abbreviation

Comment 14

Page 11. Figure 8A, 8B, 8C  “Y” axis is not informative, author is advised to elaborate the axis labelling.

Comment 15

Page 11. Line 422 E. coli and S. aureus should be italics 

Comments for author

Dear Author

The manuscript “Preparation and enhanced antimicrobial activity of thymol immobilized on different silica nanoparticles with application in apple juice” “Coatings-1639022” is required to address the following queries

The manuscript emphasize on the thymol immobilization on different silica support systems to improve the bactericidal activity for food preservative particularly apple juice.

Comment 1

This approach is converting “thymol” a GRAS material as per FDA to non GRAS HMSNs formulation, how author will justify it?

Comment 2

Immobilization approach is utilizing, Silica and CTAB which are considered to be toxic “Silica can cause silicosis, whereas CTAB is well known surfactant that can cause protein precipitation and aggregation hence regular consumptions of these two materials in lesser quantities could be harmful, what is author’s opinion about that?  

Comment 3

In the abstract section

Replace the sentence ....the results revealed (from line 23 to 25) .....but also in food system

With

The results revealed that thymol immobilized hollow mesoporus silica particles showed the...... fold higher bactericidal activity against gram negative E. coli and .....fold higher activity against gram positive S. Aureus bacteria  as compared to free thymol.

Comment 4

Page 2. Line 53, 63 –follow the same pattern for reference insertion [ either in bracket or without bracket] 

Comment 5.

Page 2. Line 74  Write the scientist name with et al instead the 6 reported (Ruiz-Rico et al reported )

Comment 6.

Page 2. Line 74-77, reframe the sentence it’s not meaningful in the current from

Comment 7.

Page 3. Section 2.2.2

Include the units of measurements (wt/wt or wt/v or v/v) in the methodology sections of manuscripts

Comment 8.

Page 5. Figure 1. Reaction scheme 1 and 2 is not clear ..insert a high resolution reaction scheme

Comment 9.

Page 6. Line 249

This may be due to TTESPC, Recheck the TTESPC... is it TTEPIC?..Author is advised to go through the entire manuscript carefully for typological errors

Comment 10.

Page 6. Line 266, 267 -Not agree with authors statement, absence of C-O-C peaks could also be result of absence of reaction of thymol and TEPIC at step 1 so include the FT-IR spectra of TTEPIC in figure 3 which will be helpful for comparative analysis of bare HMNS and immobilized HMNS with TTEPIC.

Comment 11.

Page 7. Figure 4 A, include the TTEPIC TGA spectra for comparative analysis

Comment 12.

Page 10. Figure 7. Abbreviation “I” before concentration i.e. I0.05, I0.1,IO.2 is misleading and concentrations is appears to be ten point zero five and so on ....author is advised to change the abbreviation  in figure and figure legend and elsewhere in the manuscript

Comment 13.

Page 11. Figure 8. Abbreviation “I” before concentration i.e. I0.05, I0.1,IO.2 in figure legends is misleading author s advised to change the abbreviation

Comment 14

Page 11. Figure 8A, 8B, 8C  “Y” axis is not informative, author is advised to elaborate the axis labelling.

Comment 15

Page 11. Line 422 E. coli and S. aureus should be italics 

Author Response

Thank you for your letter and for the reviewers’ comments concerning our manuscript entitled "Preparation and enhanced antimicrobial activity of thymol derivatives immobilized on different silica nanoparticles with application in apple juice". Those comments are all valuable and very helpful for revising and improving our paper, as well as the important guiding significance to our researches. We have studied comments carefully and have made correction which we hope meet with approval. Revised portion are marked in yellow in the paper. The main corrections in the paper and responses to the reviewers’ comments are in the attachment.

Author Response 

Point 1: This approach is converting “thymol” a GRAS material as per FDA to non GRAS HMSNs formulation, how author will justify it?

 Response 1: It has been reported that mesoporous silica MCM-41 was used as filter material in apple juice sterilization and the related toxicity study was also reported [1, 2].

  1. Verdu, S.; Ruiz-Rico, M.;  Perez, A. J.;  Barat, J. M.;  Talens, P.; Grau, R., Toxicological implications of amplifying the antibacterial activity of gallic acid by immobilisation on silica particles: A study on C. elegans. Environ. Toxicol. Pharmacol. 2020, 80, 103492.
  2. Peña-Gómez, N.; Ruiz-Rico, M.;  Fernández-Segovia, I.; Barat, J. M., Study of apple juice preservation by filtration through silica microparticles functionalised with essential oil components. Food Control 2019, 106, 106749.

Point 2: Immobilization approach is utilizing, Silica and CTAB which are considered to be toxic “Silica can cause silicosis, whereas CTAB is well known surfactant that can cause protein precipitation and aggregation hence regular consumptions of these two materials in lesser quantities could be harmful, what is author’s opinion about that?

Response 2: CTAB has been almost eliminated in the synthesis and the material is not directly ingested into the body. Related toxicity studies have also been reported above.

Point 3: Replace the sentence ....the results revealed (from line 23 to 25) .....but also in food system With

The results revealed that thymol immobilized hollow mesoporus silica particles showed the...... fold higher bactericidal activity against gram negative E. coli and .....fold higher activity against gram positive S. Aureus bacteria  as compared to free thymol.

Response 3: We have revised this part according to the Reviewer’s suggestion.

Point 4: Page 2. Line 53, 63 –follow the same pattern for reference insertion [ either in bracket or without bracket]

Response 4: We have revised this part according to the Reviewer’s suggestion.

Point 5: Line 74  Write the scientist name with et al instead the 6 reported (Ruiz-Rico et al reported )

Response 5: We have revised this part according to the Reviewer’s suggestion.

Point 6: Page 2. Line 74-77, reframe the sentence it’s not meaningful in the current from

Response 6: We have revised this part according to the Reviewer’s suggestion.

Point 7: Page 3. Section 2.2.2

Include the units of measurements (wt/wt or wt/v or v/v) in the methodology sections of manuscripts

Response 7: We have revised this part according to the Reviewer’s suggestion.

Point 8: Page 5. Figure 1. Reaction scheme 1 and 2 is not clear ..insert a high resolution reaction scheme

Response 8: We have revised this part according to the Reviewer’s suggestion.

Point 9: Page 6. Line 249

This may be due to TTESPC, Recheck the TTESPC... is it TTEPIC?..Author is advised to go through the entire manuscript carefully for typological errors

Response 9: We have revised this part according to the Reviewer’s suggestion.

Point 10: Page 6. Line 266, 267 -Not agree with authors statement, absence of C-O-C peaks could also be result of absence of reaction of thymol and TEPIC at step 1 so include the FT-IR spectra of TTEPIC in figure 3 which will be helpful for comparative analysis of bare HMNS and immobilized HMNS with TTEPIC.

Response 10: We have revised this part according to the Reviewer’s suggestion.

Point 11: Page 7. Figure 4 A, include the TTEPIC TGA spectra for comparative analysis

Point 11: We have revised this part according to the Reviewer’s suggestion.

Point 12: Page 10. Figure 7. Abbreviation “I” before concentration i.e. I0.05, I0.1,IO.2 is misleading and concentrations is appears to be ten point zero five and so on ....author is advised to change the abbreviation  in figure and figure legend and elsewhere in the manuscript

Point 12: We have revised this part according to the Reviewer’s suggestion.

Point 13: Page 11. Figure 8. Abbreviation “I” before concentration i.e. I0.05, I0.1,IO.2 in figure legends is misleading author s advised to change the abbreviation

Point 13: We have revised this part according to the Reviewer’s suggestion.

 Point 14: Page 11. Figure 8A, 8B, 8C  “Y” axis is not informative, author is advised to elaborate the axis labelling.

Point 14: We have revised this part according to the Reviewer’s suggestion.

Point 15: Page 11. Line 422 E. coli and S. aureus should be italics

Point 15: We have revised this part according to the Reviewer’s suggestion.

Round 2

Reviewer 1 Report

Authors addressed comments of reviewers. However, a revision is still necessary. Please follow the comments below:

L317-391 Please revise the MIC and MBC values. They are not correct. Revise this section. Figure 6 shows, for E. coli, a MIC of 0.3 mg/mL for free thymol
(A), 0.05 mg/mL for functionalized HMSNs (B), 0.1 mg/mL for MCM-41 (C) and 0.2 mg/mL for AS (D) and, for S. aureus, a MIC of 0.5 mg/mL for free thymol (A), 0.1 mg/mL for functionalized HMSNs (B), 0.1 mg/mL for MCM-41 (C) and 0.3 mg/mL for AS (D). These values can be equal to the MBC if the viability test is negative (no growth after re-inoculation on broth medium). If the viability test is positive the MBC values will be
higher than MIC. Please revise this section according to these results and add the MIC and MBC values for each compound. Thanks  

L397-399 The concentration used is a sub-lethal concentration (see figure 6). Please revise this sentence

Please discuss the molecular interactions between thymol derivatives and silica. Authors stated that "Thymol was immobilized on the supports by covalent bonds, and the functionalized supports killed bacteria in the medium by contacting the bacteria rather than releasing thymol". Please discuss this aspect in the Discussion section.

Author Response

Thank you for your letter and for the reviewers’ comments concerning our manuscript . Those comments are all valuable and very helpful for revising and improving our paper, as well as the important guiding significance to our researches. We have studied comments carefully and have made correction which we hope meet with approval. Revised portion are marked in yellow in the paper. The main corrections in the paper and responses to the reviewers’ comments are in the attachment.
